# Ethnicity/caste and child anthropometric outcomes in India using the National Family Heath Survey 2015–16 and 2019–21

**Sakshi Pandey** [1,2]*, **Dil Bahadur Rahut** [2], **Tetsuya Araki** [1]

1 Graduate School of Agricultural and Life Sciences, The University of Tokyo, Tokyo, Japan, 2 Asian Development Bank Institute, Tokyo, Japan

* pandey-sakshi@g.ecc.u-tokyo.ac.jp

**Data Availability Statement:** All relevant (completely anonymized) data supporting the conclusions of this article are available within the manuscript and its Supporting information files.

## Abstract

Socioeconomic inequalities are known to negatively impact anthropometric outcomes among children, particularly in developing countries. This study, therefore, assesses the gap in anthropometric outcomes of children 6–59 months along the ethnicity-based social groups in India using the National Family Heath Survey 2015–16 and 2019–21. The paper utilizes logistic regression models, the exogenous switching treatment effect regression (ESTER) model, and the Blinder-Oaxaca Model to disentangle the role of ethnicity (referred to as caste in India) in influencing child anthropometric outcomes while accounting for socio/ economic factors. Approximately 35% of children in the sample were stunted and 20% wasted. Result indicates that despite the progress made in reducing child undernutrition between the two survey periods, there is a higher risk of chronic growth faltering (stunting) and underweight in socially disadvantageous groups, and these ethnicity-based disparities exist independent of education and household economic status. To improve children's nutritional status, India needs to develop new nutrition strategies prioritizing double-duty action due to the persistence of undernutrition and rising overweight/obesity among children. The study suggests a need for a distinguished understanding of the underlying causes of chronic and acute forms of malnourishment, and separate interventions are required to reduce the disparities among disadvantaged groups, particularly in tribal communities.

## 1. Introduction

Health and education are two of the most critical factors in human development. Ensuring optimum health and nutritional outcomes for young children is crucial as it severely affects human capital formation and future economic prosperity [1, 2]. Childhood nutritional outcomes are widely measured through anthropometric indicators of stunting (low height-for-age), wasting (low weight-for-height), underweight (low weight-for-age), and overweight or obesity. According to the latest round of Joint Global Estimates (2022), around 148.1 million children under 5 were stunted, 45 million wasted, and 37 million are overweight. Substantive progress has been made in reducing child malnutrition—stunting prevalence declined from

**Funding:** The author(s) received no specific funding for this work.

**Competing interests:** The authors have declared that no competing interests exist.

**Abbreviations:** ESTER, Exogenous Switching Treatment Effect Regression; OBC, Other Backward Caste; SC, Scheduled Caste; SD, Socially Deprived; ST, Scheduled Tribe.

33% in 2000 to 22.3% in 2022, wasting from 8% to 6.8%, and underweight from 21% to 12.3% [3]- the statistics reveal that the world is far from achieving the 2025 World Health Assembly (WHA) global nutrition targets and the 2030 Sustainable Development Goal (SDG) 2 targets [4, 5].

Over the past two decades, India has made considerable improvement in reducing childhood undernutrition, with stunting prevalence decreasing from 48% in 2005–06 to 35.5% in 2019–21, wasting from 23% to 19.3%, and underweight prevalence from 40% to 32.1% [6]. Further, recent trends present an interesting paradox of the co-existence of a high level of undernutrition burden while an increasing prevalence of overweight among young children [7]. Existing research indicates several clinical, socioeconomic, and environment-related factors influence childhood nutrition- and health-related outcomes in India [8]. A strong correlation has been reported for household economic status, place of residence, parents' education (especially the mother), maternal anthropometric indicators such as height, weight, and age, and child feeding practices [8–10]. However, poverty is one of the most critical factors that negatively influences health and nutritional outcomes, creating a vicious and intergenerational cycle of malnutrition [11, 12].

Interestingly, poverty in India is deeply conjoined with ethnicity (referred to as caste in India), a system of social differentiation based on heredity and endogamy. According to the official classification, there are four major categories: Scheduled Tribes (STs), Scheduled Castes (SCs), Other Backward Castes (OBCs), and General/Other ("Forward") Castes. The SC and ST communities comprise about 16.6% and 8.6%, respectively, of India's population and have been colloquially referred to as 'lower or marginalized' while the rest are termed as 'upper castes' [13, 14]. A study exploring income disparity along these ethnic groups from 1961–2012 reported that SC and ST households earn 21% and 34% lower than the national average income, respectively [15]. According to the Multidimensional Poverty Index Report of 2021 produced by the United Nations Development Programme (UNDP) and the Oxford Poverty and Human Development Initiative, STs are the poorest communities in India, followed by SCs [16].

Even though caste-based discrimination is illegal in India, long-running social segregation has been associated with socioeconomic disadvantages for communities belonging to SC/STs ethnic groups. Research has further examined disparities between SC/STs and other ethnic groups in using clean energy sources and public health services, including maternal health care, schooling years, and other indicators of socioeconomic development [17–19]. Ethnicity-based inequalities trickle down further to diet qualities and health and nutritional outcomes. A study on the Consumer Expenditure Survey (CES, 2011–12) conducted by the National Sample Survey Organization (NSSO) of India pointed out that households from marginalized groups (ST/SC) consumed significantly fewer fruits and vegetables than their counterparts [20, 21]. Notably, another study assessing diet diversity among Indian children further pointed out the strong association between parental education and economic status with the intake of adequately diversified diets [22]. The overall poor performance on SES indicators has directly impacted the health and well-being of SC/ST communities, particularly children [23–25]. Consequently, the child malnutrition burdens are exceptionally high in SC and ST communities in India. Research has shown that the prevalence of stunting, anemia, and infant mortality is higher among children from socially marginalized castes (ST/SC), linking it further to a lower use of health services and uptake of iron and vitamin supplementation [23, 26–28].

A recent study using NFHS 2015–16 and Blinder-Oaxaca decomposition found that despite a decrease in under-five childhood malnutrition levels, scheduled caste children continued to show higher undernutrition levels [29]. A working paper suggested that the difference in childhood stunting incidence across marginalized caste groups (scheduled caste and tribe) is

significantly high [30]. Nonetheless, current research lacks robust methodology to explicitly identify the role of ethnicity/caste in determining childhood anthropometric outcomes by comparing them across survey rounds while also accounting for other exogenous and endogenous determinants in India.

Against this backdrop, this study aims to identify the role of ethnic groups (caste) on child anthropometric outcomes. The study explicitly explores caste-differentiated nutritional outcomes using the two consecutive rounds of the National Family Health Survey (NFHS) 2015–16 and 2019–21. Further, it uses two noble techniques: (a) Exogenous Switching Treatment Effect Regression (ESTER) and (b) the Blinder-Oaxaca decomposition model to examine the gap in child nutritional outcomes between disadvantaged ethnic groups and forward ethnic groups. Based on the assumption that ethnicity-based differences are primarily because of poverty and low economic status, the study explores the hypothesis whether a child from a socially disadvantaged group, but a higher income group and better education status is nutritionally better than a child from an upper ethnic group but lower income and education levels. The findings of this study provide an opportunity to compare the gaps and improvements between the two surveys. They may contribute to developing policies and interventions focusing on the ethnic perspective to improve child health and nutrition in the country.

## 2. Data & methodology

### 2.1. Data & Sampling

For the scope of examining malnutrition in early childhood, the research draws data on Indian children from 6 months to under 5 years of age from the two consecutive rounds of the National Family Health Survey of 2015–16 (NFHS-4) and 2019–21 (NFHS-5), conducted by the Ministry of Health and Family Welfare and the International Institute for Population Sciences, Mumbai. The NFHS 4 & 5 are cross-sectional and nationally representative surveys based on stratified multistage random sampling procedures that collect data in 29 States and 7 Union Territories of India (Jammu &Kashmir was counted as a state during NFHS-4 survey, but later recognized as a union territory by Government of India in October 2019. Hence, it was reported as a Union Territory for NFHS-5) through four survey questionnaires: the Household Questionnaire, Woman's Questionnaire, Man's Questionnaire, and Biomarker Questionnaire. Both the survey rounds adopt a uniform sample design, field procedures, biomarker measurements, and a questionnaire (translated into regional languages) to facilitate comparability across both rounds.

The NFHS-4 collected information on 221,858 children aged six months to 5 years, while NFHS-5 included information on 198,475 children nationwide. Information on children's anthropometric indicators, household demographic and socioeconomic indicators, and maternal characteristics was utilized for both the NFHS rounds. The woman's questionnaire contains information on all the children born within five years of the survey year for each participant household. After dropping the missing values for the relevant variables for this study and cleaning the dataset, the study included an effective sample 106,466 children from NFHS 2015–16 and 110,627 children from NFHS 2019–21, from 6 months to under 5 years, hereby referred to as study participants.

### 2.2. Variable description

Anthropometric measurements are one of the most widely used and scientifically robust methods to analyze malnutrition in children at the population level. This study used parameters of anthropometric failures—stunting, wasting, underweight, and overweight as the binary outcome variable (1 –if the child is malnourished, 0—otherwise). The predictor variables used in

**Table 1. Description of outcome and predictor variables used in the study.**

| S. No. | Variables | Definition/Measurement | Reference |
|---|---|---|---|
| **Outcome Variable** | | | |
| 1. | Stunting | Stunting = 1 if height-for-age z-score (HAZ) < 2 standard deviations of the WHO standard, and Stunting = 0 if otherwise. | |
| 2. | Wasting | Wasting = 1 if weight-for-height z-score (WHZ) < 2 standard deviations of the WHO standard and Wasting = 0 if otherwise. | |
| 3. | Underweight | Underweight = 1 if height-for-age z-score (HAZ) < 2 standard deviations of the WHO standard, and Underweight = 0 if otherwise. | |
| 4. | Overweight | Overweight = 1 if weight-for-height z-score (WHZ) > 2 standard deviations of the WHO standard and Overweight = 0 if otherwise. | |
| **Predictor Variable** | | | |
| 1. | Ethnicity or Caste | Coded into four categories: (i) Others or General Caste (GC), (ii) Other Backward Castes (OBC), (iii) Scheduled Castes (SC), (iv) Scheduled Tribes (ST) | [27–30] |
| 2. | Wealth Quintiles | NFHS classify the economic status of the household as 'wealth quintiles' divided into five categories: (i) poorest, (ii) poorer, (iii) middle, (iv) richer, and (v) richest. | [31–33] |
| 3. | Geographical Area | Coded as (i) Urban and (ii) Rural | [8, 34, 35] |
| 4. | Child's Age group | Coded into six categories based on the age group: (i) 6–11 months, (ii) 12–18 months, and (iii) 19–23 months, (iv) 24–35 months, (v) 36–47 months, and (vi) 48–59 months | [36, 37] |
| 5. | Child's Birthweight | Coded as 1 for children born normal birthweight (≥2.5 kg) and 0 if low birthweight (<2.5 kg) | [31, 38, 39] |
| 6. | Child's Birth order | Coded into three categories: (i) first-born, (ii) second-born, and (iii) third or later-born. | [40–42] |
| 7. | Child's Sex | Coded as 1 for male and 2 for female | [40, 41] |
| 8. | Occurrence of Infectious Disease | Coded as 1 if occurrence of fever or diarrhea in the last 15 days of the survey, 0 if otherwise | [37, 43] |
| 9. | Antenatal Care (ANC) Visits | Coded as 1 if number of visits ≥4 and 0 if number of visits <4 | [44–46] |
| 10. | Maternal Education | Coded into four categories: (i) no formal education, (ii) primary (five years of formal education), (iii) secondary (12 years of formal education), and (iv) higher (13 or more years of formal education). | [37, 38, 41, 47] |
| 11. | Maternal BMI | Coded into three categories: (i) BMI <18.5: underweight; (ii) BMI 18.5–24.9: normal weight (iii) BMI ≥25.0: overweight and obese. | [32, 41, 43] |
| 12. | Maternal Age | Coded into three categories: (i) 15–19 years, (ii) 20–35 years, and (iii) 35 years and above. | [33, 42] |
| 13. | Women as Head of HH | Coded as 1 if woman is reported as Head and 0 if otherwise | [48, 49] |
| 14. | Religion | Coded into five categories: (i) Hindu, (ii) Muslim, (iii) Sikh, (iv) Christian, and (v) Others (including Buddhist and Jain) | [25, 30, 50] |

this study are household socioeconomic characteristics, maternal characteristics, and child-level variables. The selected predictor variables were tested for multicollinearity before the analysis to ascertain the validity of the results (S1 Table). Detailed information on the outcome and predictor variables are presented in Table 1.

## 2.3. Empirical methodology

**2.3.1. Logistic model.** The study uses logistic regression analysis to understand the association of ethnicity with malnutrition in children, separately for NFHS 2015–16 and 2019–21. Regression coefficients with robust standard errors are reported for both survey rounds for stunting, wasting, underweight, and overweight. The models adjusted for the socioeconomic factors and the mother's and child's characteristics while accounting for geographical differences, thus controlling for states as a dummy variable.

**2.3.2. Interaction between ethnicity and other SES indicators—Marginal effect analysis.** To evaluate the role of ethnicity in association with other socioeconomic variables on the anthropometric outcome, the paper assessed the interaction between ethnicity and wealth index, ethnicity and maternal education level, ethnicity, and geographical location (urban vs. rural) for each anthropometric indicator in readjusted logistic models. The interaction terms

provide the opportunity to examine whether the effect of social groups on anthropometric outcome changes with the change in other SES indicators such as wealth index, maternal education level, and geographical location (urban vs. rural).

For both rounds of NFHS, marginal effects with robust standard errors are reported for each interaction-effect readjusted model. Marginal effect is a partial derivative from a regression and provide unit-specific and sample average predictions from models. It allows to interpret the magnitude of the effects of the independent variable on the outcome variable rather than the direction of changes.

### 2.3.3. Exogenous switching treatment effect regression (ESTER).

While providing insight into the direction and magnitude of the relationship between ethnicity and anthropometric failures in children, logistic regressions are limited to providing the intercept effect (i.e., parallel shift effect or homogenous slope hypothesis) that does not change with other covariates affecting anthropometric outcomes in children. Existing studies exploring causality due to differentiation between two groups have demonstrated the capacity of ESTER models [51–53]. ESTER models allow the estimation of counterfactual outcomes, comparing two groups regarding the impact of a treatment or intervention on an outcome variable when the treatment assignment is exogenous. Therefore, this study utilizes the ESTER model to evaluate the causal impact of caste-based ethnicity on children's anthropometric outcomes through a counterfactual framework.

Based on the distribution of anthropometric failure wrt ethnicity of each study participant, the ESTER employs a counterfactual framework, with separate equations for socially deprived (SC and ST) ethnic groups (sd = 1) and other ethnic groups (OBCs and General Caste) (sd = 0). The model can be estimated as follows:

$$y_{sd} = X_{sd}\beta_{sd} + U_{sd} \; if \; sd = 1 \qquad 1(a)$$

$$y_{oc} = X_{oc}\beta_{oc} + U_{oc} \; if \; sd = 0 \qquad 1(b)$$

In Eqs 1(a) and 1(b), subscript $sd$ represents socially deprived ethnic group belonging to SC and ST communities, and $oc$ represents other groups; $y$ represents the presence of anthropometric indicator (separate equations are derived for stunting, wasting, underweight, and overweight) by each group based on the subscript; $X$ is the explanatory variable vector; $\beta$ is the observed magnitude of effect for children belonging to each category; and $U$ is the random error assumed to be of constant variance and zero mean.

Following the ESTER framework established by Carter and Milon, 2005 [54] and employed in existing socioeconomic studies [53, 55], the counterfactual model is explained by the following equations:

$$E(y_{sd} \,|sd = 1) = X_{sd}\beta_{sd} \qquad 2(a)$$

$$E(y_{oc} \,|sd = 0) = X_{oc}\beta_{oc} \qquad 2(b)$$

$$E(y_{oc} \,|sd = 1) = X_{sd}\beta_{oc} \qquad 2(c)$$

$$E(y_{sd} \,| sd = 0) = X_{oc}\beta_{sd} \qquad 2(d)$$

Eqs 2(a) and 2(b) represent the actual condition for anthropometric outcomes for children in socially deprived and other ethnic households. Eqs 2(c) and 2(d) represent respective counterfactual scenarios that show the probability of occurrence of anthropometric failure of children in one ethnic group if the returns to their response variables had the same coefficients as

**Table 2. Exogenous Switching Treatment Effects Regression (ESTER) methodology.**

|  | Socially Deprived Castes (SD) | Other Castes (OC) | Treatment Effect |
|---|---|---|---|
| Socially Deprived Ethnic Groups (SD) | (a) $E(y_{sd}\|sg = 1)$ | (c) $E(y_{oc}\| sg = 1)$ | TT = (a)–(c) |
| Other Castes (OC) | (d) $E(y_{sd}\|sg = 0)$ | (b) $E(y_{oc}\| sg = 0)$ | TU = (d)–(b) |
| Heterogeneity effect (difference caused by unobserved characteristics) | $HE_{sd}$ = (a)–(d) | $HE_{oc}$ = (c)–(b) |  |

the other ethnic group, and vice versa. Table 2 further explains in detail the methodology employed by ESTER in determining the treatment and heterogeneity effects.

If the child's characteristics from the SD community had the same coefficients as from the OC community characteristics returns, the effect of ethnicity on the SD child's anthropometric outcome (TT) could be given by the difference between Eqs 2(a) and 2(b), as shown in Eq (3):

$$TT = E(y_{sd} \mid sd = 1) - E(y_{oc} \mid sd = 1) = X_{sd}(\beta_{sd} - \beta_{oc}) \tag{3}$$

Analogously, the effect of ethnicity on the child from an OC Community (TU)–if their characteristics had the same returns (coefficients) as SD's characteristics return—is given by the difference between Eqs 2(c) and 2(d):

$$TU = E(y_{sd} \mid sd = 0) - E(y_{oc} \mid sd = 0) = X_{oc}(\beta_{sd} - \beta_{oc}) \tag{4}$$

However, there can also be a possibility that the child from an SD community may intrinsically have better outcomes than her counterpart in an OC Community irrespective of their observed characteristics because of other endogenous determinants and vice-versa. In such a case, the difference can be represented as Eqs 2(a) and 2(c) and Eqs 2(b) and 2(d), as outlined in Eqs (5) and (6):

$$HE_{sd} = E(y_{sd} \mid sd = 1) - E(y_{sd} \mid sd = 0) = \beta_{sd}(X_{sd} - X_{oc}) \tag{5}$$

$$HE_{oc} = E(y_{sd} \mid sd = 1) - E(y_{sd} \mid sd = 0) = \beta_{oc}(X_{sd} - X_{oc}) \tag{6}$$

**2.3.4. Blinder—Oaxaca model.** The Blinder-Oaxaca decomposition model divides the outcome differential between two groups into two parts [56, 57]. The first part of the model consists of variations that can be explained by group differences through economic and other determinants. The second part consists of differences that cannot be directly accounted for through independent variables. For linear regression [58], the standard Blinder-Oaxaca decomposition of the gap between two groups for an outcome Y can be presented as:

$$\bar{y}^a - \bar{y}^b = \left[\left(\bar{X}^a - \bar{X}^b\right)\beta^a\right] + \left[\bar{X}^b\left(\beta^a - \beta^b\right)\right] \tag{7}$$

$\bar{X}^j$ is a row vector of average values of the independent explanatory variables and $\beta^j$ is the estimation coefficient for group $j$. However, since the outcome variable considered in this study (anthropometric failure in children) is binary, this standard method cannot be used directly. Utilizing the approach by existing studies [59, 60], the Blinder-Oaxaca decomposition for a logistic model can be represented as:

$$\bar{y}^a - \bar{y}^b = \sum_{i=1}^{N^a} \frac{F(X_i^a\beta^a)}{N^a} - \sum_{i=1}^{N^b} \frac{F(X_i^b\beta^a)}{N^b} + \sum_{i=1}^{N^b} \frac{F(X_i^b\beta^a)}{N^b} - \sum_{i=1}^{N^b} \frac{F(X_i^b\beta^b)}{N^b} \tag{8}$$

## 3. Results & discussion

### 3.1. Descriptive statistics

The sample showed that the prevalence of stunting was highest among the three anthropometric outcomes, with almost 35% of the children stunted. In NFHS 2015–16 sample, 35.83% (n = 38,149) children were stunted, while in NFHS 2019–21, 34.85% (n = 38,548) children under 5 were stunted. For wasting, 20% of children were observed to be wasted, with the prevalence slightly decreasing from 20.6% (n = 21,921) in NFHS 2015–16 to 18.5% (n = 20,446) in 2019–21. Approximately 30% of children were underweight in the combined sample, again with an overall decrement from 32.6% (n = 34,675) in NFHS-4 to 28.3% (n = 31,346) in NFHS-5. The statistics suggest the overall prevalence of overweight is around 3% but with a slight increase from 2.5% (n = 2623) in NFHS-4 to 3.6% (n = 3969) in NFHS-5.

Table 3 shows the distribution of study participants by NFHS rounds for all the explanatory variables used in this study. Almost 40% of the children included in the study belonged to socially backward (SC & ST) communities, with 78% reporting Hinduism as the religion. In terms of gender, the study participants were approximately equally distributed, with 46% female and 54% male participants. Regarding birth order, more than 70% of children in the sample were either first or second-born for both NFHS rounds, with more than half being two years or older. Further, 80% of the study participants were born with a normal birthweight (2.5 kgs or more) for two NFHS rounds of 2015–16 to 2019–21.

In terms of mother's characteristics, more than half of mothers had an education above the secondary level; however, only 14.5% had attained higher education, with a marginal improvement from 13% in 2015–16 to 16% in 2019–21. Further, only 13% of the households reported having the head as female. In the overall sample, 58% of the mothers received 4 or more ANC visits, slightly improving from 55% in 2015–16 to 61% in 2019–21.

### 3.2. Disaggregated descriptive analysis by Socioeconomic Status (SES)

Several studies assessing the children's nutritional outcomes in India and other South Asian economies have reported a strong association between socioeconomic status (SES) indicators and anthropometric failures [17, 37, 42]. Fig (1a)–(1h) graphically present the distribution of each anthropometric failure with the SES indicators used in the study separately for each NFHS round. Fig (1a) and (1b) suggest that stunting, wasting, and underweight prevalence were highest for SC category study participants, followed by ST, OBC, and the general category for both rounds of NFHS 2015–16 and 2019–21. However, the prevalence of overweight among the participants was similar across all the ethnic categories.

On the other hand, in Fig (1c) and (1d), the prevalence of stunting, wasting, and being underweight decreased as the wealth quintile moved up, contrary to overweight, which increased with the wealth quintile. A similar trend was also observed for maternal education levels in the NFHS rounds. Regarding the malnutrition burden in urban vs. rural India, Fig (1g) and (1h) suggest that the overall burden is comparatively higher in rural areas, except for overweight. These preliminary findings are coherent with the existing studies that have pointed out the unequal distribution of malnutrition burden among the different socioeconomic groups, with a particular concern in rural areas. The findings from the descriptive analysis set the context for this study, paving the way for a higher-level analysis to determine the association of malnutrition with SES indicators.

**Table 3. Distribution of study participants (6–59 months) with each explanatory variable.**

| | NFHS-4 (2015–16) | NFHS-5 (2019–21) |
|---|---|---|
| | % | % |
| **Age (in months)** | | |
| 6–11 months | 14.7 | 13.6 |
| 12–17 months | 15.1 | 14.7 |
| 18–23 months | 14.3 | 12.8 |
| 24–35 months | 22.7 | 23.1 |
| 36–47 months | 18.4 | 19.1 |
| 48–59 months | 14.8 | 16.7 |
| **Sex of Child** | | |
| Male | 54.8 | 54 |
| Female | 45.2 | 46 |
| **Birth Order** | | |
| 1st | 35.2 | 33.8 |
| 2nd | 35.4 | 37 |
| 3rd or more | 29.5 | 29.2 |
| **Normal birthweight (2.5 kgs or more)** | 83.7 | 83.8 |
| **4 or more ANC Visits of Mother** | 55.2 | 61.4 |
| **BMI of Mother** | | |
| Underweight (BMI<18.5) | 23.0 | 18.5 |
| Normal (BMI≥18.5 & <24.5) | 60.1 | 61.2 |
| Overweight & Obese (BMI≥25) | 16.9 | 20.3 |
| **Education of Mother** | | |
| No Formal Education | 22.1 | 18.3 |
| Primary Education | 13.3 | 11.8 |
| Secondary Education | 51.6 | 53.9 |
| Higher Education | 12.9 | 16.0 |
| **Area of the Household** | | |
| Rural | 28.6 | 22.1 |
| Urban | 71.4 | 77.9 |
| **Households with Women as Head** | 11.8 | 14.9 |
| **Wealth Quintile** | | |
| Poorest | 18.9 | 22.5 |
| Poorer | 20.9 | 22.7 |
| Middle | 21.3 | 20.5 |
| Richer | 20.2 | 18.6 |
| Richest | 18.6 | 15.6 |
| **Ethnic Group** | | |
| General | 20.7 | 18.2 |
| Other Backward Castes | 41.8 | 40.6 |
| Scheduled Castes | 19.3 | 20.9 |
| Scheduled Tribes | 18.3 | 20.4 |
| **Religion** | | |
| Hindu | 77.8 | 77.7 |
| Muslim | 10.7 | 10.7 |
| Christian | 2.3 | 2.0 |
| Sikh | 7 | 7.4 |
| Other | 2.2 | 2.3 |

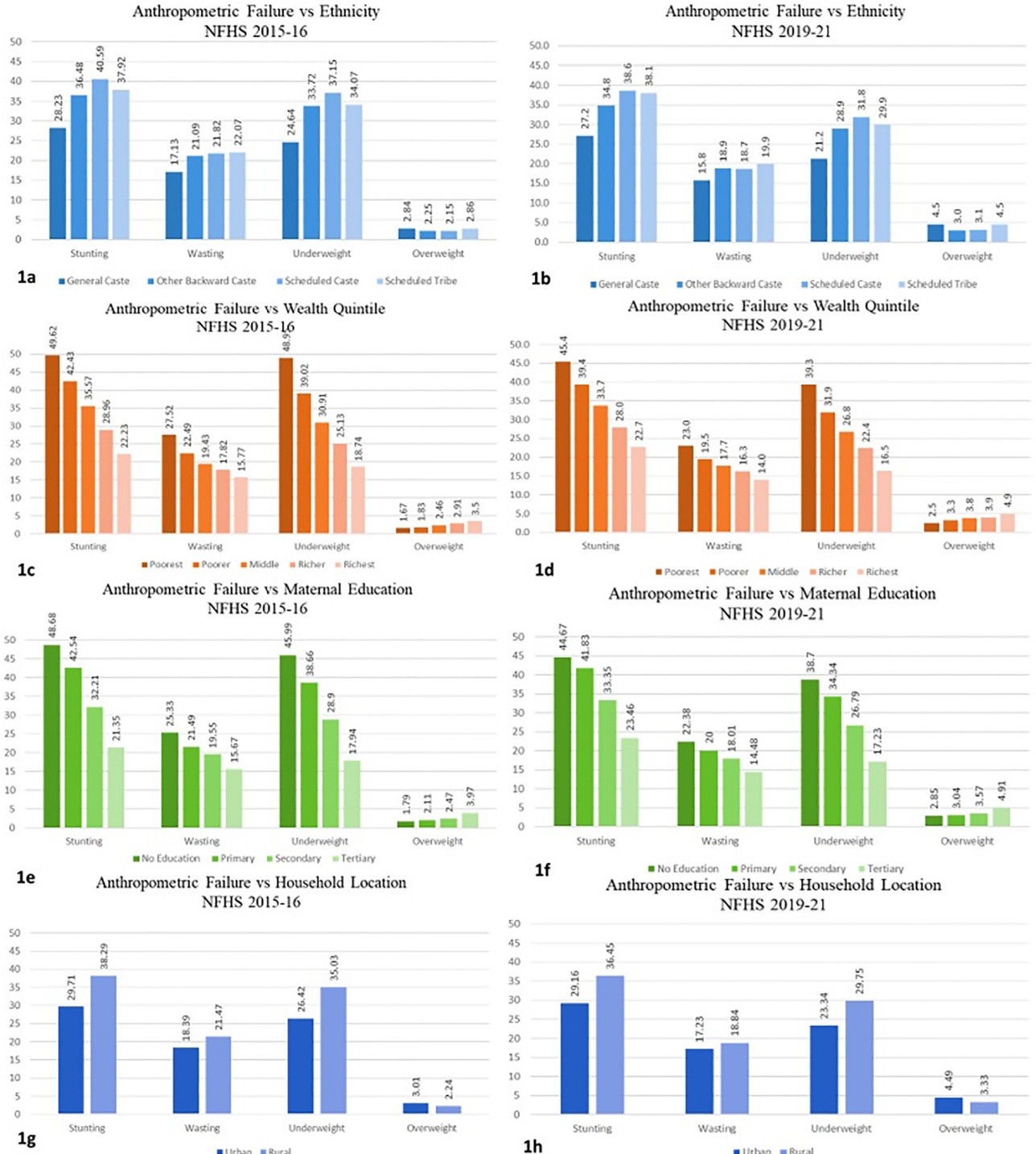

**Fig 1. Prevalence of anthropometric failure in study participants (6–59 months) from NFHS 2015–16 and 2019–21 with selected SES indicators.**

### 3.3 Empirical results & discussion

**3.3.1. Factors influencing child's anthropometric failure (logistic model).** Tables 4 and 5 present the Odds Ratios (ORs) for the prevalence of stunting, wasting, underweight, and overweight in children (6–59 months) using the logistic regression model for NFHS-4 (2015–16) and NFHS-5 (2019–21), respectively.

*3.3.1.1. Ethnicity/Caste.* As suggested from the preliminary descriptive analysis, the logistic results for NFHS 2015–16 (Table 4) and 2019–21 (Table 5) show a significant and negative association with ethnic groups, with children from ST and SC households being approximately 1.3 times at risk of stunting, wasting, and underweight than their General Caste and OBC counterparts. However, regarding overweight, a non-significant association was observed for ethnic groups.

*3.3.1.2. Wealth Index.* Further, as expected, the results reported a strong association of anthropometric outcomes with the wealth index. The risk of stunting is 0.4 to 0.5 times in the wealthiest quintile compared to the poorest for both NFHS 2015–16 and 2019–21. Similarly, for wasting, the risk was 0.6 times for participants from the wealthiest household in NFHS-4 and 0.7 times in NFHS-5. Assessment of underweight also showed a similar trend, with risk-reducing to 0.4 times in both NFHS-4 and NFHS-5. Contrary to the trends, it was observed in the NFHS 2015–16 dataset that participants from the wealthiest quintile were 1.5 times more at risk of being overweight, with the risk being higher by 1.7 times in 2019–21.

*3.3.1.3. Maternal Education Level.* The analysis observed a similar trend for association between maternal education level and anthropometric indicators. Higher maternal education levels significantly reduced the risk of stunting and underweight in participants for both rounds of surveys. However, no significant association was observed for overweight. The risk was reduced to approximately 0.7 times for stunting and 0.6 times for underweight in participants with mothers with higher education for both NFHS rounds. Further, for wasting, a slight reduction was observed between participants whose mothers attained higher education compared to mothers without education.

*3.3.1.4. Location of the household (urban vs rural).* Contrary to the disaggregated descriptive analysis results, the logistic regression analysis suggests a reverse association between urbanicity and anthropometric failure when other factors were controlled. The ORs for all four anthropometric indicators suggested that participants in rural areas have a slightly lower or similar risk of suffering from anthropometric failures than urban study participants. The result suggests that the household's location has no exclusive role in affecting anthropometric outcomes, and the higher prevalence of undernutrition in rural areas can be attributed to other socioeconomic and clinical factors.

*3.3.1.5. Child's Characteristics.* The logistic results indicate a significant association between participants with a birth weight of more than 2.5 kg and all the anthropometric outcomes for NFHS 2014–15 and 2019–21. The risk was observed to be reduced to 0.7 for stunting, 0.8 times for wasting, and 0.6 times for underweight. As apparent, there was a slight increment in the risk of being overweight for children born with a weight of more than 2.5 kg. Moreover, the birth order of the child was also observed to harm anthropometric outcomes, with second and third-born children being at higher risk of stunting, wasting, and being underweight when compared to first-born children. Further, fever or diarrhea within two weeks of the survey significantly increased the risk of wasting and underweight, as wasting measures acute undernourishment, and underweight is a composite indicator.

The participant's age group was also observed to have a significant association with all the anthropometric outcomes. The ORs suggested that the risk of stunting and being underweight increased with age group and was highest for children of 18–23 months, with a gradually

**Table 4. Odds ratio and robust standard errors for anthropometric failures in study participants (6–59 months) for NFHS (2015–16).**

| Variables | Stunting | Wasting | Underweight | Overweight |
|---|---|---|---|---|
| **Household Ethnicity** | | | | |
| *General (base category)* | | | | |
| Other Backward Castes | 1.155(0.023)*** | 1.049(0.024)*** | 1.133(0.023)*** | 0.898(0.052)* |
| Scheduled Castes | 1.305(0.03)*** | 1.068(0.029)*** | 1.251(0.03)*** | 0.868(0.059)** |
| Scheduled Tribes | 1.221(0.033)*** | 1.25(0.038)*** | 1.303(0.036)*** | 0.942(0.073) |
| **Wealth Index** | | | | |
| *Poorest (base category)* | | | | |
| Poorer | 0.877(0.019)*** | 0.86(0.021)*** | 0.813(0.017)*** | 0.938(0.072) |
| Middle | 0.719(0.017)*** | 0.765(0.021)*** | 0.627(0.015)*** | 1.089(0.085) |
| Richer | 0.575(0.015)*** | 0.717(0.022)*** | 0.492(0.013)*** | 1.229(0.103)*** |
| Richest | 0.456(0.014)*** | 0.663(0.024)*** | 0.371(0.012)*** | 1.419(0.13)*** |
| **Area of the Household** | | | | |
| *Urban (base category)* | | | | |
| Rural | 0.961(0.017)** | 0.921(0.019)*** | 0.905(0.017)*** | 0.947(0.046) |
| **Maternal Education Level** | | | | |
| *No Education (base category)* | | | | |
| Primary | 0.916(0.021)*** | 0.893(0.024)*** | 0.897(0.021)*** | 1.018(0.08) |
| Secondary | 0.77(0.015)*** | 0.931(0.021)*** | 0.791(0.016)*** | 0.931(0.06) |
| Higher | 0.597(0.018)*** | 0.843(0.029)*** | 0.6(0.019)*** | 1.222(0.101)*** |
| **Gender** | | | | |
| *Male (base category)* | | | | |
| Female | 0.886(0.012)*** | 0.878(0.014)*** | 0.919(0.013)*** | 0.938(0.038) |
| **Occurrence of fever or Diarrhea** | 0.973(0.016) | 1.037(0.02)** | 1.06(0.018)*** | 0.779(0.042)*** |
| **Birthweight more than 2.5 kg** | 0.657(0.012)*** | 0.718(0.014)*** | 0.548(0.01)*** | 1.327(0.082)*** |
| **BMI of Mother** | | | | |
| *<18.5 (base category)* | | | | |
| ≥18.5 & <25 | 0.889(0.012)*** | 0.76(0.014)*** | 0.639(0.01)*** | 1.764(0.111)*** |
| ≥25 | 0.657(0.012)*** | 0.498(0.014)*** | 0.43(0.011)*** | 1.918(0.145)*** |
| **4 or more visits to ANC** | 0.937(0.014)*** | 1.02(0.018) | 0.975(0.015) | 0.934(0.043) |
| **Gender of the head** | | | | |
| *Male (base category)* | | | | |
| Female | 0.998(0.021) | 0.921(0.023)*** | 0.96(0.021)* | 1.036(0.064) |
| **Mother's age group** | | | | |
| *15–19 years (base category)* | | | | |
| 20–34 years | 0.947(0.043) | 1.107(0.056)** | 0.969(0.045) | 1.036(0.144) |
| 35 years or older | 0.925(0.048) | 1.155(0.068)** | 0.955(0.05) | 1.261(0.195) |
| **Birth Order** | | | | |
| *1st (base category)* | Reference | Reference | Reference | Reference |
| 2nd | 1.106(0.018)*** | 1.001(0.019) | 1.084(0.019)*** | 0.855(0.04)*** |
| 3rd or later | 1.206(0.023)*** | 0.985(0.022) | 1.137(0.022)*** | 0.76(0.044)*** |
| **Age Group** | | | | |
| *6–11 months (base category)* | | | | |
| 12–17 Months | 2.068(0.054)*** | 0.847(0.023)*** | 1.261(0.033)*** | 0.696(0.045*** |
| 18–23 Months | 2.972(0.078)*** | 0.718(0.02)*** | 1.589(0.042)*** | 0.538(0.038)*** |
| 24–35 Months | 2.424(0.059)*** | 0.79(0.02)*** | 1.761(0.042)*** | 0.43(0.028)*** |
| 36–47 Months | 2.447(0.063)*** | 0.722(0.019)*** | 1.793(0.046)*** | 0.488(0.032)*** |

(*Continued*)

**Table 4.** (Continued)

| Variables | Stunting | Wasting | Underweight | Overweight |
|---|---|---|---|---|
| | **Odds Ratio & Robust Standard Errors** | | | |
| 48–59 Months | 2.046(0.056)*** | 0.656(0.019)*** | 1.699(0.046)*** | 0.574(0.039)*** |
| **Religion** | | | | |
| Hindu *(base category)* | | | | |
| Muslim | 1.135(0.026)*** | 1.032(0.028) | 1.059(0.026)** | 0.908(0.067) |
| Sikhs | 0.788(0.054)*** | 1.038(0.085) | 0.83(0.062)** | 0.785(0.144) |
| Christian | 0.94(0.042) | 0.953(0.053) | 0.928(0.046) | 1.059(0.111) |
| Others | 1.047(0.052) | 1.039(0.062) | 1.042(0.057) | 0.999(0.126) |
| **Constant** | 0.574 (0.090)*** | 0.708(0.112)*** | 0.647(0.094)*** | 0.036(0.012)*** |

***p value <0.01,

** p-value <0.05;

*p value <0.1

[model was controlled for the locational effects using state dummies; however, coefficients for the state dummies are not presented due to space constraints]

decreased risk in later age groups. Stunting represents chronic undernourishment in children and may not be present in the initial months but gradually manifest as the child ages [2, 61]. The risk for stunting starts manifesting as the child grows up and becomes 3 times around the age of 2, pointing to the importance of early childhood care and proper implementation of IYCF practices, including initiating nutritious food through complementary feeding to meet the growing needs of children. Underweight is a composite indicator of acute and chronic undernourishment, showing a similar trend. On the other hand, as expected, wasting showed the opposite trend because as the risk of being stunted increased and the child is at risk of being shorter, there is a lower risk of being wasted.

*3.3.1.6. Mother's Characteristics.* The ORs for the logistic regression indicate that the mother's BMI significantly impacted the anthropometric outcomes for both NFHS 2015–16 and 2019–21. Apparently, children whose mother had a normal BMI were at lower risk of being undernourished. However, the risk of overweight was significantly higher in participants with mothers being obese compared to those with normal BMI. Interestingly, the ORs for NFHS 2019–21 suggest that the children whose mothers were 20 years or older at birth had lower chances of being stunted and underweight. ORs for NFHS 2015–16 also showed a similar association, however insignificant. The results are coherent with the earlier studies and indicate the importance of women's education and late marriages concerning pregnancy and child health [48, 62]. Moreover, association with ANC visits were incoherent and non-significant for all four anthropometric measurements.

*3.3.1.7. Other Household-level Characteristics.* For both NFHS 2015–16 and 2019–21, the ORs suggest that the gender of the Household Head was only significantly associated with wasting outcome, with the risk being slightly reduced if the head was a woman. Further, religion also was observed to have no significant association with anthropometric outcomes. For the NFHS 2015–16 analysis, a modest improvement in stunting risk was observed for Sikh households; however, the association was non-significant for 2019–21. Similarly, a slight increase in risk for stunting was observed for Muslim households in 2019–21 but non-significant in 2015–16.

**3.3.2. Interaction of ethnicity/caste with SES indicators—Marginal effect analysis.** Social inequality due to ethnicity/caste has consistently been associated with poverty, poor educational status, and a higher prevalence in rural areas. Based on the initial results from

**Table 5. Odds ratio and robust standard errors for anthropometric failures in study participants (6–59 months) for NFHS (2019–21).**

| Variables | Stunting | Wasting | Underweight | Overweight |
|---|---|---|---|---|
| | Odds Ratio & Robust Standard Errors | | | |
| **Household Ethnicity** | | | | |
| General *(base category)* | | | | |
| Other Backward Castes | 1.159(0.024)*** | 1.1(0.027)*** | 1.153(0.026)*** | 0.874(0.043)*** |
| Scheduled Castes | 1.303(0.03)*** | 1.122(0.032)*** | 1.294(0.032)*** | 0.914(0.05) |
| Scheduled Tribes | 1.242(0.033)*** | 1.259(0.04)*** | 1.285(0.037)*** | 0.963(0.061) |
| **Wealth Index** | | | | |
| Poorest *(base category)* | | | | |
| Poorer | 0.865(0.017)*** | 0.901(0.021)*** | 0.848(0.017)*** | 1.187(0.067)*** |
| Middle | 0.729(0.016)*** | 0.849(0.022)*** | 0.721(0.017)*** | 1.284(0.077)*** |
| Richer | 0.593(0.015)*** | 0.804(0.024)*** | 0.614(0.016)*** | 1.271(0.083)*** |
| Richest | 0.504(0.015)*** | 0.776(0.029)*** | 0.49(0.016)*** | 1.564(0.116)*** |
| **Area of the Household** | | | | |
| Urban *(base category)* | | | | |
| Rural | 0.962(0.018)** | 0.867(0.02)*** | 0.909(0.019)*** | 0.906(0.039)** |
| **Maternal Education Level** | | | | |
| No Education *(base category)* | | | | |
| Primary | 0.981(0.023)* | 0.955(0.027) | 0.958(0.024)* | 0.963(0.065) |
| Secondary | 0.84(0.016)*** | 0.906(0.021)*** | 0.826(0.017)*** | 0.908(0.049)* |
| Higher | 0.686(0.019)*** | 0.831(0.028)*** | 0.664(0.02)*** | 1.069(0.071) |
| **Sex** | | | | |
| Male *(base category)* | | | | |
| Female | 0.864(0.011)*** | 0.88(0.014)*** | 0.884(0.012)*** | 0.984(0.032) |
| **Occurrence of Fever or Diarrhea** | 0.98(0.016) | 1.042(0.021)** | 1.063(0.019)*** | 0.832(0.038)*** |
| **Birthweight more than 2.5 kg** | 0.671(0.012)*** | 0.758(0.015)*** | 0.579(0.01)*** | 1.172(0.058)*** |
| **BMI of Mother** | | | | |
| <18.5 *(base category)* | | | | |
| $\geq$18.5 & <25 | 0.805(0.014)*** | 0.803(0.016)*** | 0.665(0.012)*** | 1.454(0.079)*** |
| $\geq$25 | 0.665(0.015)*** | 0.558(0.015)*** | 0.458(0.011)*** | 1.758(0.108)*** |
| **4 or more visits to ANC** | 0.985(0.014) | 1.056(0.019)*** | 1.007(0.015) | 1.02(0.038) |
| **Gender of the head** | | | | |
| Male *(base category)* | | | | |
| Female | 1.013(0.019) | 0.922(0.021)*** | 0.955(0.019)** | 0.947(0.046) |
| **Mother's age group** | | | | |
| 15–19 years *(base category)* | | | | |
| 20–34 years | 0.907(0.045) | 1.065(0.06) | 0.88(0.045)*** | 0.943(0.111) |
| 35 years or older | 0.817(0.044)*** | 1.085(0.069) | 0.804(0.046)*** | 1.138(0.147) |
| **Birth Order** | | | | |
| 1st *(base category)* | | | | |
| 2nd | 1.129(0.018)*** | 1.06(0.021)*** | 1.158(0.02)*** | 0.854(0.033)*** |
| 3rd or later | 1.266(0.023)*** | 1.048(0.024)** | 1.287(0.026)*** | 0.766(0.037)*** |
| **Age Group** | | | | |
| 6–11 months *(base category)* | | | | |
| 12–17 Months | 1.765(0.046)* | 0.931(0.026)** | 1.285(0.035)*** | 0.649(0.036)*** |
| 18–23 Months | 2.556(0.067)*** | 0.822(0.025)*** | 1.642(0.046)*** | 0.562(0.034)*** |
| 24–35 Months | 1.915(0.046)*** | 0.911(0.024)*** | 1.71(0.043)*** | 0.429(0.023)*** |
| 36–47 Months | 1.96(0.049)*** | 0.765(0.021)*** | 1.71(0.045)*** | 0.516(0.028)*** |

*(Continued)*

**Table 5.** (Continued)

| Variables | Odds Ratio & Robust Standard Errors | | | |
| --- | --- | --- | --- | --- |
| | Stunting | Wasting | Underweight | Overweight |
| 48–59 Months | 1.59(0.042)*** | 0.771(0.023)*** | 1.686(0.046)*** | 0.67(0.036)*** |
| **Religion** | | | | |
| Hindu *(base category)* | | | | |
| Muslim | 1.108(0.027)*** | 1.157(0.033)*** | 1.137(0.029)*** | 0.999(0.06) |
| Sikhs | 0.874(0.064)* | 1.058(0.106) | 1.007(0.087) | 1.221(0.175) |
| Christian | 1.003(0.042) | 1.112(0.058)** | 1.004(0.048) | 1.081(0.094) |
| Others | 1.025(0.052) | 0.941(0.06) | 1.049(0.06) | 1.471(0.138)*** |
| **Constant** | 0.466(0.038)*** | 0.488(0.044)*** | 0.598(0.052)*** | 0.099(0.017)*** |

***p value <0.01,

** p-value <0.05;

*p value <0.1

[model was controlled for the locational effects using state dummies; however, coefficients for the state dummies are not presented due to space constraints]

descriptive statistics and logistic regression suggesting a significant detrimental effect of caste/ethnicity on child anthropometric outcomes, the study further explores whether the detrimental effect is exclusive because of the participant's ethnicity or a combined effect of poverty, low education, or urbanicity. The study uses a logistic regression model that applies interaction between ethnicity and each of the other three SES indicators (exclusive for each SES indicator) while controlling for all other variables, including states. Each regression model was analyzed separately for NFHS 2015–16 and 2019–21. Marginal effects were then calculated for interactions, and coefficients were plotted for each anthropometric indicator and both rounds of NFHS.

*3.3.2.1. Interaction of Ethnicity/Caste with Wealth Index.* Fig (2a)–(2h) represent the coefficient for marginal effect analysis for determining the role of ethnicity in interaction with the wealth index associated with stunting, wasting, underweight, and overweight outcomes in children from 6 months to under 5 years of age using NFHS 2015–16 and 2019–21 data. The marginal effects for stunting show that the risk is not significantly different among ethnic groups in the poorest and wealthiest quintiles. However, in the middle three wealth index categories, children from the general caste (GC) were at lower risk than their counterparts. For wasting, the coefficients suggest that risk was almost constant along the wealth quintiles for GC children. However, it was decreasing for children from other ethnic groups. Interestingly, in the wealthiest quintile, all the children were observed to have equal risk despite their ethnicity. Like stunting, underweight also showed a decreased risk as children moved along the wealth quintile, with a minimal difference between children of the wealthiest quintile. Marginal effects for overweight individuals did not show a clear trend with wealth index and ethnicity; however, it was observed that the risk of being overweight increased with the wealth quintile for all ethnicities.

*3.3.2.2. Interaction of Ethnicity/Caste with Maternal Education.* Fig (3a)–(3h) show the marginal effects of interaction between maternal education level and ethnicity of study participants for all four anthropometric failures, separately for NFHS 2015–16 and 2019–21. Fig (3a) and (3b) show that stunting risk decreases for all ethnicities when the maternal education level increases. For the highest education level, children from all ethnic groups except GC have a similar risk of stunting. However, for lower education levels, SC and ST children were observed to have a higher risk than their counterparts. The marginal effects of wasting did not

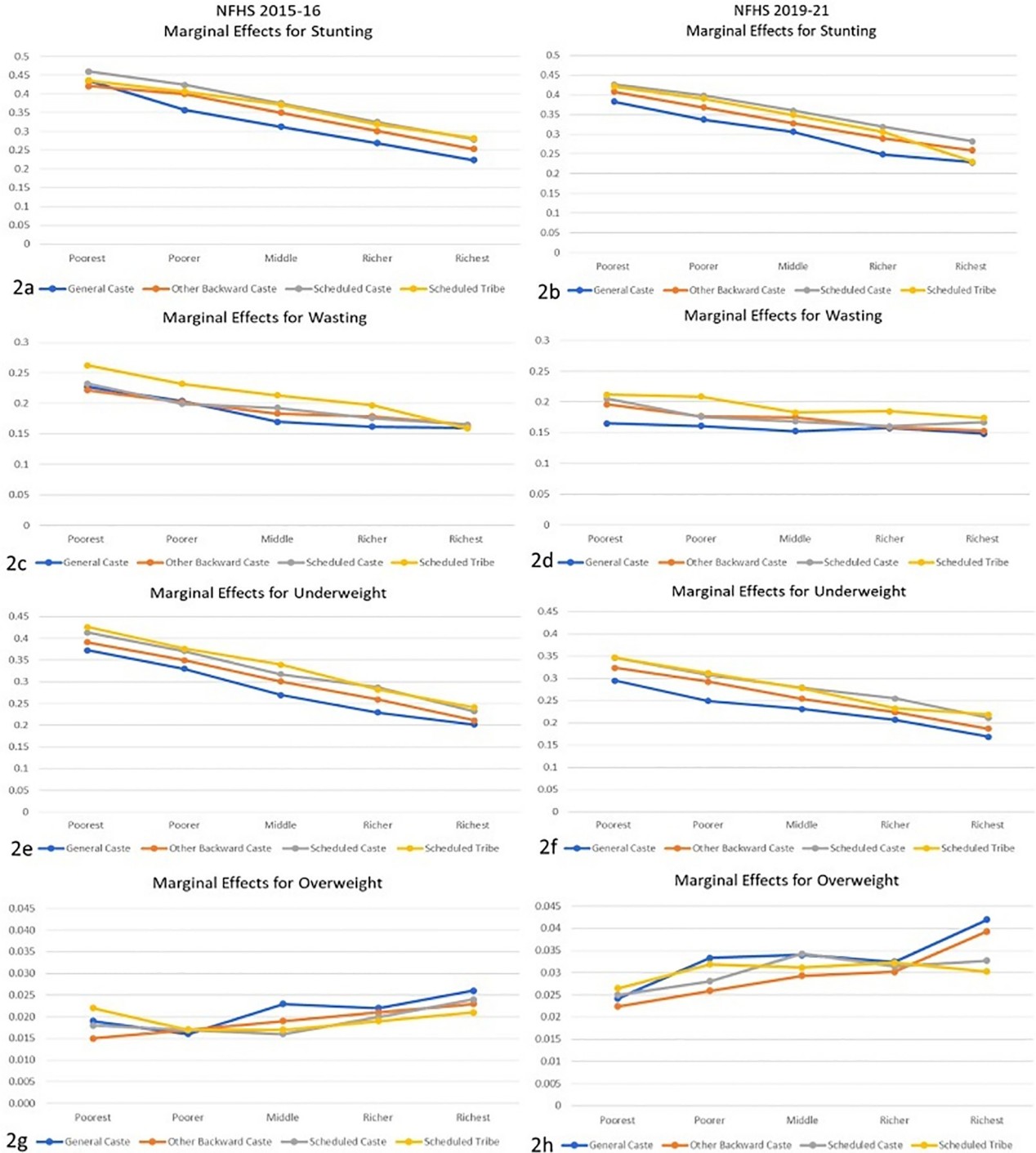

**Fig 2. Marginal effects for anthropometric failures in study participants from logistic regression with interaction between ethnicity/case and wealth index.**

show a clear trend with maternal education level. However, it was observed that children from the ST community were at higher risk at each education level in both NFHS-4 and 5. In Fig (3e) and (3f), the risk declined for underweight as the maternal education level increased for all the ethnic categories. The risk among children from SC and ST communities declined

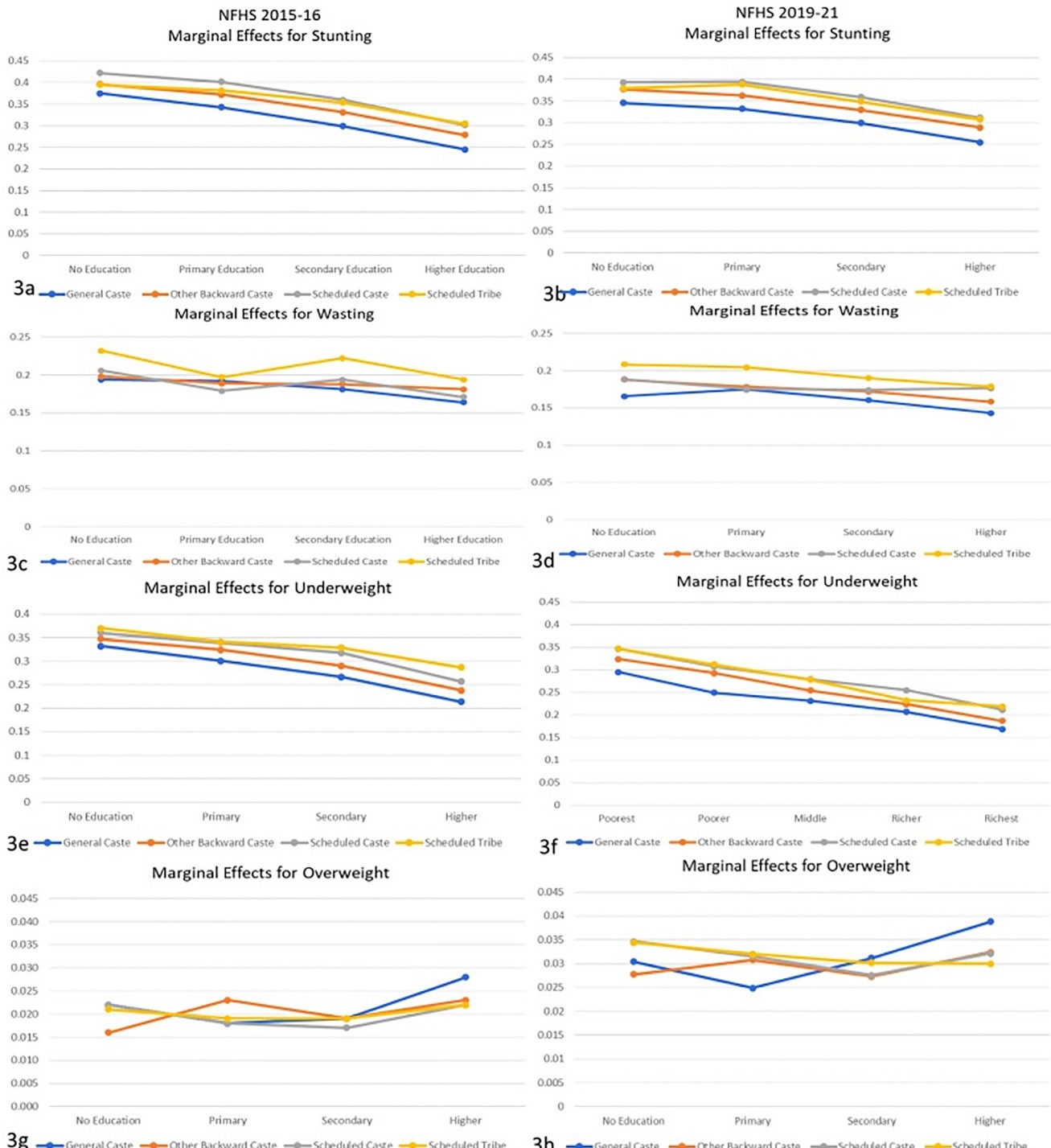

**Fig 3. Marginal effects for anthropometric failures in study participants from logistic regression with interaction between ethnicity/caste and education level of the mother.**

steeply with education level; however, they were still at a slightly higher risk than OBC and GC children. Trends for overweight show that the risk increases if a mother has an education level above primary. However, at no education level, SC and ST children had a higher risk of being overweight, while at the highest education level, GC children were at the highest risk.

*3.3.2.3. Interaction of Ethnicity/Caste with Location of the Household.* Fig (4a)–(4h) represent marginal effects for anthropometric failures using logistic regression with interaction between ethnicity and location of the household. Fig (4a) and (4b) represent the marginal effects of stunting for NFHS 2015–16 and 2019–21, respectively. The trend shows that children in rural areas have a slightly lower risk than in urban areas. The risk among OBC, SC, and ST children remained similar in urban and rural areas. However, an evident difference between the ethnic groups was observed, with GC children at lower risk in urban and rural areas. Fig (4c) and (4d) graphically present marginal effects for wasting outcomes. The trend shows that wasting risk was the same for all the ethnicities in urban areas for both rounds of the survey, and except for ST children, it decreased in rural areas. For children from the ST community, NFHS 2015–16 showed a slight increment in risk, while 2019–21 showed no improvement. Fig (4e) and (4f) represent underweight marginal effects and show a similar trend to stunting. Underweight risk nominally decreased for rural children, with GC children having the least risk among all the children, followed by OBC, SC, and ST children. Fig (4g) and (4h) represent the marginal effects of overweight risk. Notably, the risk of being overweight was higher in urban areas than in rural areas, with GC children at a higher risk than their counterparts.

In conclusion, the overall marginal effect analysis for household location contradicts the prevailing argument that rural areas are a more critical hotspot of child undernutrition than urban areas. The analysis points out that if other conditions are controlled, children from rural areas are better off than their urban contemporaries.

**3.3.3. Ethnicity and anthropometric failure—Exogenous Switching Treatment Effects Regressions (ESTER).** As the core aim of this research is to determine the influence of ethnic groups on the prevalence of anthropometric failures, the study created two distinct categories of ethnicity for ESTER analysis. Based on the disaggregated descriptive statistics and logistic regression result, it was observed that SC and ST children were at a significantly higher risk for all the indicators than OBC and GC children. For the analysis, GC and OBC children were categorized into the base category of Other Caste (OC) and a comparison was made with ST and SC children, who were categorized into the Socially Deprived (SD) group.

*3.3.3.1. ESTER analysis for stunting.* Table 6 summarizes the results of the exogenous switching treatment effect regression (ESTER) model, which unpacks the impact of ethnic identity (SC-ST vs. General and OBC) on the stunting outcome in children from 6–59 months for NFHS 2015–16 and 2019–21 survey. The results show that the difference in the stunting outcome between the Socially Deprived (SD) group (ST-SC) and Others (General and OBC) is determined by both the treatment effect and heterogeneity effects. The results suggest the gap between SCs-STs and other ethnic groups children concerning the stunting outcome is primarily because of the heterogeneous economic and other characteristics. However, the potential disadvantage that the community experiences in Indian society further exacerbates the gap.

*3.3.3.2. ESTER Analysis for Wasting.* Table 7 represents the ESTER model results for wasting outcomes in the study participants. The treatment and heterogeneity effect sizes were minimal, particularly for NFHS-5; however, they were significant. The result indicates that wasting outcomes are affected both by ethnicity and other heterogeneous differences in other characteristics. However, the gap is reduced when comparing the results between NFHS-4 and 5.

*3.3.3.3. ESTER Analysis for Underweight.* Table 8 summarizes the result of the ESTER analysis for underweight outcomes in children. Similar to stunting and wasting, the higher risk

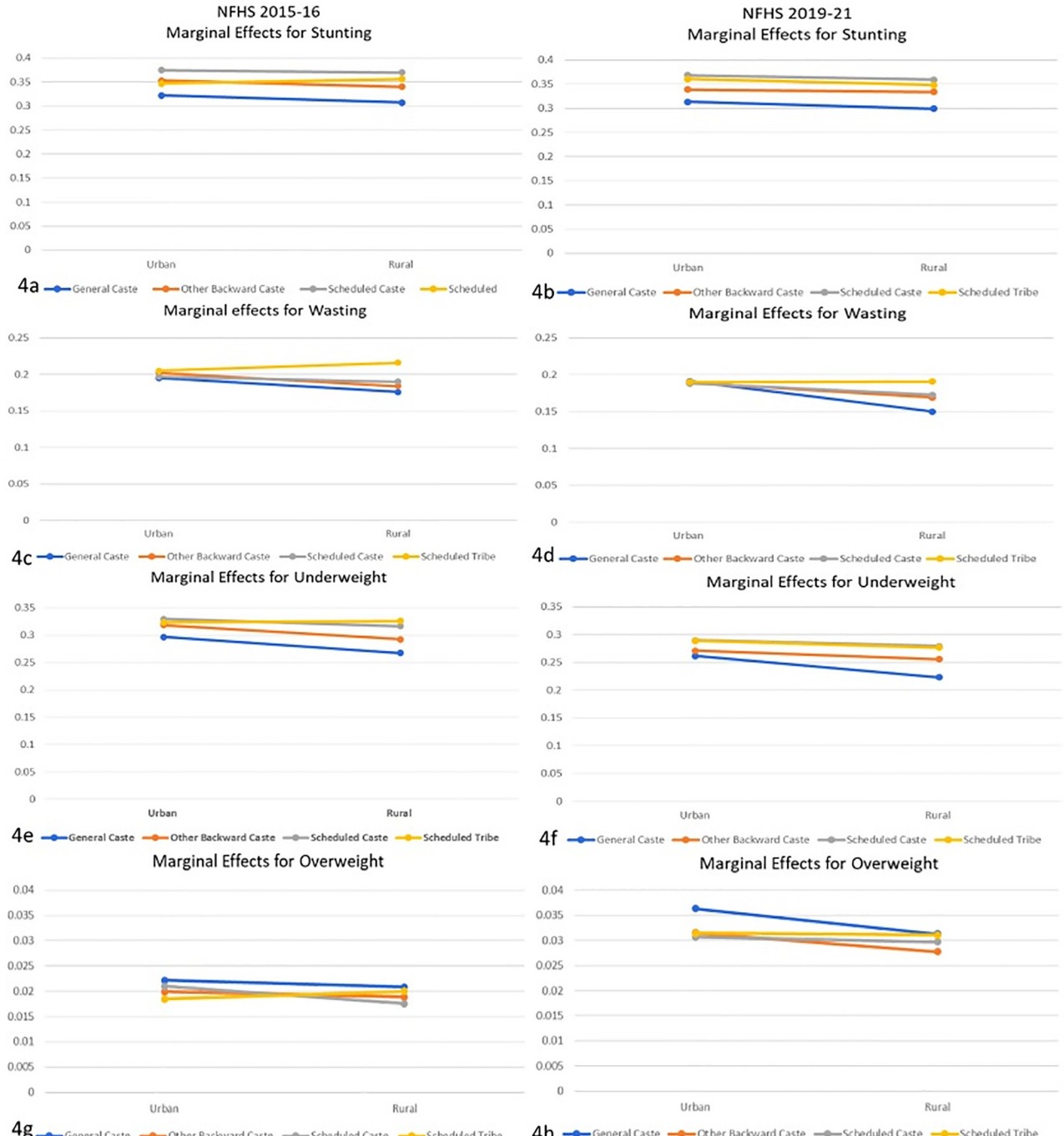

**Fig 4. Marginal effects for anthropometric failures in study participants from logistic regression with interaction between ethnicity/caste and geographical location of the household.**

among SC-ST children can be attributed both to differences in socioeconomic, clinical, and other determinants and the hierarchal disadvantage faced by the community in India. Underweight results also indicate that in the period between 2015–16 and 2019–21, the difference due to hierarchy has reduced marginally.

**Table 6. Exogenous switching treatment effect regression (ESTER) for stunting prevalence in study participants from NFHS 2015–16 and NFHS 2019–21.**

| | NFHS-4 (2015–16) | | |
|---|---|---|---|
| Ethnic Group | Socially Deprived Castes (SD) | Other Castes (OC) | Treatment Effect |
| Socially Deprived (SD) (ST-SC) | 0.393 [a] (.0007) | 0.37 [c] (0.0005) | 0.023***(0.0009) |
| Other Castes (OC) | 0.358[d] (0.0008) | 0.338[b] (0.0005) | -0.021***(0.0009) |
| Heterogeneity effect | 0.035*** (0.0001) | -0.032*** (0.0007) | 0.055*** (0.0009) |
| | NFHS-5 (2019–21) | | |
| Ethnic Group | Socially Deprived Castes (SD) | Other Castes (OC) | Treatment Effect |
| Socially Deprived (SD) (ST-SC) | 0.383 [a] (0.0006) | 0.352 [c] (0.0005) | 0.032***(0.0007) |
| Other Castes (OC) | 0.351 [d] (0.0006) | 0.324 [b] (0.0005) | -0.027***(0.0007) |
| Heterogeneity effect | 0.032***(0.0008) | -0.028*** (0.0007) | 0.059*** (0.0007) |

***p value <0.01,

** p-value <0.05;

*p value <0.1

*3.3.3.4. ESTER Analysis for Overweight.* Table 9 summarizes the ESTER model results for overweight outcomes in study children. Results from NFHS-4 were insignificant for both treatment and heterogeneity effects, thus not suggesting any difference due to ethnicity or other socioeconomic differences. However, in NFHS 2019–21, the treatment and heterogeneity effects were significant, and the effect sizes were small. The overall results for all the anthropometric failures indicate that despite the constitutional provisions of equality, SC-STs still face disadvantages, and thus, social reformation is essential to alleviate the residual effects of traditional ethnic divisions.

**3.3.4. Robustness check using Blinder-Oaxaca model.** The study further employed the Blinder-Oaxaca model to check the robustness of ESTER results. The result for NFHS 2015–16 in Table 10 shows a 5.5% gap between the SC-ST and other children, 3.8% due to endowments, 1.1% due to coefficients, and the interaction term contributes 0.6%. A 2.2% gap was observed in the risk of wasting between the two ethnic groups. 1.5% of the above difference was explained by endowments, 1% by coefficients, and the interaction effect reduced the gap by about 0.4% (significant at p-value <0.1). Similarly, a 4.9% difference was observed for the

**Table 7. Exogenous switching treatment effect regression (ESTER) for wasting prevalence in study participants from NFHS 2015–16 and 2019–21.**

| | NFHS-4 (2015–16) | | |
|---|---|---|---|
| Ethnic Group | Socially Deprived Castes (SD) | Other Castes (OC) | Treatment Effect |
| Socially Deprived (SD) (ST-SC) | 0.219 [a] (0.0005) | 0.21 [c] (0.0003) | 0.009***(0.0005) |
| Other Castes (OC) | 0.211 [d] (0.0004) | 0.198 [b] (0.0003) | -0.012***(0.0005) |
| Heterogeneity effect | 0.009*** (0.0006) | -0.012*** (0.0004) | 0.022*** (0.0005) |
| | NFHS-5 (2019–21) | | |
| Ethnic Group | Socially Deprived Castes (SD) | Other Castes (OC) | Treatment Effect |
| Socially Deprived (SD) (ST-SC) | 0.193 [a] (0.0003) | 0.190 [c] (0.0003) | 0.003***(0.0004) |
| Other Castes (OC) | 0.190 [d] (0.0003) | 0.179 [b] (0.0002) | -0.01*** (0.0004) |
| Heterogeneity effect | 0.003*** (0.0005) | -0.01*** (0.0004) | 0.014*** (0.0004) |

***p value <0.01,

** p-value <0.05;

*p value <0.1

**Table 8. Exogenous switching treatment effect regression (ESTER) for underweight prevalence in study participants from NFHS 2015–16 and 2019–21.**

| | NFHS-4 (2015–16) | | |
| --- | --- | --- | --- |
| Ethnic Group | Socially Deprived Castes (SD) | Other Castes (OC) | Treatment Effect |
| Socially Deprived (SD) (ST-SC) | 0.356 [a] (0.0009) | 0.337 [c] (0.0005) | 0.019*** (0.001) |
| Other Castes (OC) | 0.349 [d] (0.0008) | 0.307 [b] (0.0006) | -0.041*** (0.0009) |
| Heterogeneity effect | 0.08*** (0.0012) | -0.03*** (0.0008) | 0.059*** (0.001) |
| | NFHS-5 (2019–21) | | |
| Ethnic Group | Socially Deprived Castes (SD) | Other Castes (OC) | Treatment Effect |
| Socially Deprived (SD) (ST-SC) | 0.309 [a] (0.0006) | 0.294 [c] (0.0005) | 0.014*** (0.0008) |
| Other Castes (OC) | 0.288 [d] (0.0006) | 0.265 [b] (0.0005) | -0.023*** (0.0007) |
| Heterogeneity effect | 0.021*** (0.0009) | 0.029*** (0.0007) | 0.043*** (0.0008) |

***p value <0.01,

** p-value <0.05;

*p value <0.1

underweight risk, of which endowments explained 3.5%, 0.7% by coefficients, and 0.7% by interaction. No significant difference was observed between the two ethnic groups in terms of the probability of overweight outcomes in the study participants.

Table 11 summarizes the result of Blinder-Oaxaca decomposition for NFHS 2019–21. Regarding stunting, a 5.9% variation was observed between the children from SC-ST and other communities. 3.7% of the difference was explained by endowments, 1.6% by coefficients, and 0.6% by interaction effect. A gap of 1.4% was observed for wasting risk, with 1.3% explained by endowments and 0.6% by interaction with coefficients, reducing the gap by 0.6% (significant at p-value <0.05). For the risk of underweight, a 4.3% gap was observed between the two ethnic groups. 3% of the difference was explained by endowments, 1.3% by interaction, and 0.1% by coefficients (p-value>0.1). Summarizing the result for overweight, the model observed a 0.3% gap, of which 0.1% is contributed by endowments, 0.7% by coefficients (p-value>0.1), and reduced by 0.4% by interaction term. The analysis overall confirms the disadvantageous position of children from SC-ST communities, particularly for stunting and

**Table 9. Exogenous switching treatment effect regression (ESTER) for overweight prevalence in study participants from NFHS 2015–16 and 2019–21.**

| | NFHS-4 (2015–16) | | |
| --- | --- | --- | --- |
| Ethnic Group | Socially Deprived Castes (SD) | Other Castes (OC) | Treatment Effect |
| Socially Deprived (SD) (ST-SC) | 0.025 [a] (0.0001) | 0.024 [c] (0.0001) | 0.0013*** (0.0001) |
| Other Castes (OC) | 0.026 [d] (0.0001) | 0.024 [b] (0.00007) | 0.0003 (0.0001) |
| Heterogeneity effect | -0.0004* (0.0002) | -0.0013** (0.0001) | -0.0001*** (0.0001) |
| | NFHS-5 (2019–21) | | |
| Ethnic Group | Socially Deprived Castes (SD) | Other Castes (OC) | Treatment Effect |
| Socially Deprived (SD) (ST-SC) | 0.038 [a] (0.0001) | 0.033 [c] (0.000) | 0.005*** (0.0001) |
| Other Castes (OC) | 0.040 [d] (0.0003) | 0.034 [b] (0.0001) | -0.005*** (0.0002) |
| Heterogeneity effect | -0.002*** (0.0003) | 0.002*** (0.0001) | 0.004*** (0.0002) |

***p value <0.01,

** p-value <0.05;

*p value <0.1

**Table 10. Anthropometric failure difference in study participants between ethnic groups using Blinder Oaxaca model for NFHS 2015–16.**

|  | Stunting | Wasting | Underweight | Overweight |
|---|---|---|---|---|
| Group 1 | 0.338*** | 0.198*** | 0.307*** | 0.024*** |
| (General-OBC) | (0.002) | (0.002) | (0.002) | (0.001) |
| Group 2 | 0.393*** | 0.219*** | 0.356*** | 0.025*** |
| (SC-ST) | (0.002) | (0.002) | (0.002) | (0.001) |
| Difference | -0.055*** | -0.022*** | -0.049*** | -0.001 |
|  | (0.003) | (0.003) | (0.003) | (0.001) |
| Endowments | -0.038*** | -0.015*** | -0.035*** | 0.0004 |
|  | (0.002) | (0.002) | (0.002) | (0.001) |
| Coefficients | -0.011*** | -0.01*** | -0.007** | -0.004*** |
|  | (0.003) | (0.003) | (0.003) | (0.001) |
| Interaction | -0.006** | 0.004* | -0.007*** | 0.003*** |
|  | (0.002) | (0.002) | (0.002) | (0.001) |

***p value <0.01,

** p-value <0.05;

*p value <0.1

underweight outcomes, that denote chronic malnourishment in children. The analysis strongly advocates that policies should pay more attention to SC-ST communities, mainly to reduce stunting and other forms of chronic failure.

## 4. Conclusion & policy recommendations

Sustainable Development Goal 2 aims to eliminate all forms of malnutrition by 2030, and more than 10 of the 17 SDGs include indicators relevant to nutrition. Moreover, the World Health Assembly in 2012 endorsed Global Nutrition Targets that strive to achieve a 40% reduction in the overall prevalence of stunting, a 5% reduction in wasting, and a 30%

**Table 11. Anthropometric failure difference in study participants between ethnic groups using Blinder Oaxaca model for NFHS 2019–21.**

|  | Stunting | Wasting | Underweight | Overweight |
|---|---|---|---|---|
| Group 1 | 0.324*** | 0.179*** | 0.266*** | 0.034*** |
| (General-OBC) | (0.002) | (0.002) | (0.002) | (0.001) |
| Group 2 | 0.383*** | 0.193*** | 0.309*** | 0.038*** |
| (SC-ST) | (0.002) | (0.002) | (0.002) | (0.001) |
| Difference | -0.059*** | -0.014*** | -0.043*** | -0.003*** |
|  | (0.003) | (0.002) | (0.003) | (0.001) |
| Endowments | -0.037*** | -0.013*** | -0.03*** | -0.001*** |
|  | (0.002) | (0.001) | (0.002) | (0.001) |
| Coefficients | -0.016*** | 0.006** | -0.001 | -0.007 |
|  | (0.003) | (0.003) | (0.003) | (0.001) |
| Interaction | -0.006** | -0.006*** | -0.013*** | 0.004*** |
|  | (0.003) | (0.002) | (0.002) | (0.001) |

***p value <0.01,

** p-value <0.05;

*p value <0.1

reduction in low birth weight by 2025. Following these targets, the Government of India launched the National Nutrition Mission or Poshan Abhiyaan in 2018 to reduce stunting, undernutrition, and low birth weight to two percent per annum. India has been putting consistent efforts in reducing malnutrition in children under 5; however, from 2005 to 2021, the decrease rate has stagnated at around one percent per annum. Studies in the past analyzing child malnutrition in the Indian context have repeatedly pointed towards socioeconomic inequalities as a predominant factor, and limited progress has been made in addressing these inequalities [37, 63].

Using unit-level data from two latest and consecutive rounds of cross-sectional National Family Health Survey 2015–16 and 2019–21, this study disentangles the role of ethnicity or caste-based social segregation on the occurrence of malnutrition in children from 6–59 months in India. The findings reveal that despite the progress made by India in reducing child undernutrition, children from SC and ST communities are more vulnerable to undernourishment as compared to OBC and GC children. Additionally, the analysis suggests that when it comes to acute anthropometric failure (wasting), the gap was minimal among the ethnic groups; however, for stunting and underweight, which represent chronic undernourishment, the difference between the ethnic groups was much higher. In summary, there is a higher risk of stunting and being underweight in socially disadvantageous groups, and these ethnicity-based disparities exist independent of education and household economic status.

The empirical results from the study further highlight that household economic status and maternal characteristics significantly affect anthropometric outcomes besides ethnicity. The marginal effect analysis noted that the risk of all anthropometric failures decreased as economic status improved. In particular, for the wealthiest quintile, all the children were observed to have equal risk for stunting and underweight despite their ethnicity. The results further highlighted that higher maternal education levels significantly reduced the risk of stunting and underweight in children. The study observed that children from ST communities were at the most risk of undernourishment, even at the same socioeconomic level. Despite the higher prevalence of undernutrition in rural areas, the findings from marginal effects analysis reflect that geographical location has no significant role in children's anthropometric outcomes. However, the high burden in rural areas can be attributed to low socioeconomic status and lack of sufficient healthcare infrastructure to ensure efficient pre- and post-natal care.

India currently has one of the highest burdens of stunting in children in Asia and the World. Stunting not only signifies physical growth faltering but has lasting effects on cognitive development and intergeneration nutritional outcomes. A World Bank report stated that "childhood stunting results in a 1.4% loss in economic productivity for every 1% loss in adult height" [64]. One of the most noteworthy findings from the study is that socioeconomic status (education, income, and ethnicity) had a more prominent impact on chronic undernourishment indicators, i.e., stunting. At the same time, while improvement in acute anthropometric failure, i.e., wasting, was observed with improvement in socioeconomic determinants, there was no apparent trend. The results indicate that in the short term, socioeconomic and social hierarchy-based disadvantages may not seem apparent; however, in the long term, they have a substantial negative impact on the overall growth and development of children.

The analytical results further show that a healthy birthweight is crucial in preventing anthropometric failures in children, particularly stunting and underweight. Although the prevalence of low birth weight has decreased from 21% in 2005–06 to 18% in 2015–16, it has stagnated at 18% for 2019–21 [7, 65]. To improve the nutritional outcomes in children, it is imperative for policies to identify predictors of low birth weight and address the issue. Moreover, the logistic regression analysis indicates that the mother's BMI or nutritional status is critical in improving the child's nutritional outcomes. Notably, the stunting risk in children

exacerbates as the child grows old and is highest for children from 18 months to 2 years of age. In summary, the findings signify that prenatal care (including the provision of nutritious supplementary food during pregnancy), continuous monitoring of children during the first five years, and proper implementation of IYCF practices of exclusive breastfeeding for the initial six months and complementary feeding later are incredibly crucial to break the intergenerational loop of malnutrition.

India needs to develop new nutrition strategies prioritizing double-duty action due to the persistence of undernutrition and rising overweight/ obesity among children. There needs to be immediate coordination between the ministries, properly juxtaposing health-related and nutrition-related programs from pregnancy to age five, and critical progress monitoring. While multiple governmental initiatives have focused on diet-based interventions, including supplementary feeding through hot-cooked meals and take-home rations [66–68], the results from the study indicate a need for a distinguished understanding of the underlying causes of chronic and acute forms of malnourishment and separate interventions. In conclusion, the study suggests that improving socioeconomic status, particularly income and education, can be pivotal in enhancing children's nutritional status, particularly for stunting, irrespective of their ethnic backgrounds. Water, Sanitation, and Hygiene (WASH) based interventions, specifically in rural areas and urban slums, can be crucial in reducing the incidence of fever and diarrheal infections [69, 70], thus preventing acute growth failures. However, since the results also show that ethnicity-based disparities exist independent of other socioeconomic determinants, specific initiatives at various levels are required to reduce the disparities among disadvantaged groups, particularly in tribal communities, irrespective of their educational and wealth status. Future research in this area could explore the potential impact of specific policy interventions by the Government of India, such as the direct cash transfer (DCT) scheme for empowering women from disadvantaged communities and Pradhan Mantri Matru Vandana Yojana (PMMVY) (maternity benefit program) through long-term panel data, pre-post analysis, experiments or randomized control trials.

## Supporting information

**S1 Table. Variance Inflation Factor (VIF) and tolerance values for predictor variables used in the study.**
(DOCX)

**S1 Data.**
(XLSX)

## Acknowledgments

The authors acknowledge the Japanese Government (Monbukagakusho) Scholarship program and the University of Tokyo for supporting the discussion.

## Author Contributions

**Conceptualization:** Sakshi Pandey, Dil Bahadur Rahut, Tetsuya Araki.

**Data curation:** Sakshi Pandey.

**Formal analysis:** Sakshi Pandey.

**Investigation:** Sakshi Pandey.

**Methodology:** Sakshi Pandey, Dil Bahadur Rahut.

**Project administration:** Tetsuya Araki.

**Resources:** Tetsuya Araki.

**Supervision:** Dil Bahadur Rahut, Tetsuya Araki.

**Validation:** Dil Bahadur Rahut, Tetsuya Araki.

**Writing – original draft:** Sakshi Pandey.

**Writing – review & editing:** Sakshi Pandey, Dil Bahadur Rahut, Tetsuya Araki.

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
