## [Decision Letter · Decision Letter 0]

3 Jul 2024

PONE-D-24-16005Understanding the impact of Ethnicity/Caste and Child Anthropometric Outcomes in India using the National Family Heath Survey 2015-16 and 2019-21PLOS ONE

Dear Dr. Pandey,

Thank you for submitting your manuscript to PLOS ONE. After careful consideration, we feel that it has merit but does not fully meet PLOS ONE’s publication criteria as it currently stands. Therefore, we invite you to submit a revised version of the manuscript that addresses the points raised during the review process.

We look forward to receiving your revised manuscript.

Kind regards,

Prem Shankar Mishra

Academic Editor

PLOS ONE

2.   In the online submission form, you indicated that your data will be submitted to a repository upon acceptance.  We strongly recommend all authors deposit their data before acceptance, as the process can be lengthy and hold up publication timelines. Please note that, though access restrictions are acceptable now, your entire minimal  dataset will need to be made freely accessible if your manuscript is accepted for publication. This policy applies to all data except where public deposition would breach compliance with the protocol approved by your research ethics board. If you are unable to adhere to our open data policy, please kindly revise your statement to explain your reasoning and we will seek the editor's input on an exemption. 

Reviewers' comments:

Reviewer's Responses to Questions

**Comments to the Author**

1. Is the manuscript technically sound, and do the data support the conclusions?

Reviewer #1: Yes

Reviewer #2: No

2. Has the statistical analysis been performed appropriately and rigorously? 

Reviewer #1: Yes

Reviewer #2: Yes

3. Have the authors made all data underlying the findings in their manuscript fully available?

Reviewer #1: No

Reviewer #2: No

4. Is the manuscript presented in an intelligible fashion and written in standard English?

Reviewer #1: Yes

Reviewer #2: Yes

5. Review Comments to the Author

Reviewer #1: 1- Clear subheadings are needed in the methodology section to differentiate between various components of the methodology.

2- comprehensive review of previous studies is lacking in the manuscript, which would give strength to justify current research.

3- Rationale behind the selection of ESTER and marginal analysis needs to be elaborated.

4- unexpected findings within the result are not deeply discussed (i.e, mother's age group at birth was insignificant for anthropometric outcomes, etc), which could be an opportunity for future research.

5- Number figure for underweight is missing in the introduction.

Reviewer #2: My Comments on

PONE-D-24-16005

Understanding the impact of Ethnicity/Caste and Child Anthropometric Outcomes in

India using the National Family Heath Survey 2015-16 and 2019-21

Since the table numbers do not match with the referred table numbers, I would suggest the authors resubmit the paper. The following points should be taken into account before resubmitting.

1. The description of variables is too big. Instead, it would be nice if the list of variables taken in the analysis is presented in the form of a table with columns as (i) name of the variable, (ii) a brief description of the variable and (iii) the categories if it is a categorical variable. The required explanations may be given as brief as possible.

2. It is wrong to say that the “logistic model fails to account for the interaction between ethnicity and other covariates”. However, the ESTER and Blinder Oaxaca model models are welcome.

3. Table numbers and referred table numbers do not match. In fact, table numbering is not OK. After Table 1 it jumps to Table 11.

4. Table 12 is not necessary.

5. While or before carrying out the logistic regression it is necessary to test for multicollinearity of the explanatory variables.

6. Because the paper is too lengthy, I would suggest removing the figures completely.

6. PLOS authors have the option to publish the peer review history of their article (what does this mean?). If published, this will include your full peer review and any attached files.

Reviewer #1: No

Reviewer #2: No

---

## [Author Response · Author response to Decision Letter 0]

14 Aug 2024

We thank the handling editor and the two anonymous reviewers for their valuable and constructive comments. We agree with most of the comments and have tried our best to address the comments raised by the reviewer. The suggestions and recommendations provided by the reviewer have helped us significantly improve the paper. We have indicated changes or additions to the main text in red in the revised manuscript with track changes. Our point-by-point responses are provided below:

Response to Reviewer 1

Comment 1- Clear subheadings are needed in the methodology section to differentiate between various components of the methodology.

Response 1: With lots of thanks for this good suggestion, we have included the sub-heading to explain the four different methodologies used in this study.

Comment 2- A comprehensive review of previous studies is lacking in the manuscript, which would give strength to justify current research. 

Response 2: Thank you very much. In the current paper, we have made a significant revision in the introduction section as well as in the results and discussion. Due to the limited availability of space, we could not create a new section devoted to the literature review; instead, we built the review in the introduction [Page 4, Line 45 to Page 6, Line 116] and throughout the manuscript. In total, we have used 71 references in this paper. 

Comment 3- Rationale behind the selection of ESTER and marginal analysis needs to be elaborated. 

Response 3: Thank you very much; we included the rationale for the selection of ESTER [Page 9, Line 171-180] and marginal analysis [Page 8, Line 158 to Page 9, Line 169]. 

Comment 4- Unexpected findings within the result are not deeply discussed (i.e, mother’s age group at birth was insignificant for anthropometric outcomes, etc), which could be an opportunity for future research. 

Response 4: Thank you very much for this suggestion. We welcome the suggestion, have revised the discussion section, and included a discussion on unexpected findings, such as the mother’s age group. [Page 17, Line 358 - 364]

Comment 5- Number figure for underweight is missing in the introduction. 

Response 5: Thank you very much for pointing out the error. We have added the number figure for underweight in the introduction section [Page 4, line 53]. 

Response to Reviewer 2

Comment 1: Since the table numbers do not match with the referred table numbers, I would suggest the authors resubmit the paper. 

Response 1: We sincerely thank the reviewer for pointing out the error in table numbering. We have revised the table numbers throughout the manuscript. 

Comment 2: The description of variables is too big. Instead, it would be nice if the list of variables taken in the analysis is presented in the form of a table with columns as (i) name of the variable, (ii) a brief description of the variable and (iii) the categories if it is a categorical variable. The required explanations may be given as brief as possible.

Response 2: Thank you for your valuable suggestion. We agree with your comment and have revised the variable description section by removing the descriptions entirely and replaced with a detailed table including information on the description of variables and citing relevant literature for the rationale to use those explanatory variables [Page 8, line 139-147]. 

Comment 3: It is wrong to say that the “logistic model fails to account for the interaction between ethnicity and other covariates”. However, the ESTER and Blinder Oaxaca model models are welcome. 

Response 3: Thank you for your suggestion. We have corrected the above statement [Page 9, line 171-174] and have added additional information on the rationale behind using ESTER [Page 9, Line 174-180].

Comment 4: Table numbers and referred table numbers do not match. In fact, table numbering is not OK. After Table 1 it jumps to Table 11. 

Response 4: Thank you for pointing this out. We have corrected the numbering throughout the revised version of the manuscript. 

Comment 5: Table 12 is not necessary. 

Response 5: We thank the reviewer. The submitted manuscript did not have Table 12, but we reckon that the reviewer might be referring to Table 2. We welcome the suggestion, given the space constraint of the journal. We have removed Table 2 in the revised manuscript and added information in the descriptive results section [Page 12, Line 240 - 248]. 

Comment 6: While or before carrying out the logistic regression it is necessary to test for multicollinearity of the explanatory variables. 

Response 6: Thank you for the critical suggestion. We analyzed the Variance Inflation Factor (VIF) for all the predictor variables before finalizing the results [Page 8, Line 144-145]. The VIF Value was equal to or less than 5 for every category of predictor variables. We have added the table indicating VIF Values and Tolerance level for each predictor variable as a supporting document at the end of the manuscript [Page 50-51, line 915]. 

Comment 7: Because the paper is too lengthy, I would suggest removing the figures completely. 

Response 7: We thank the reviewer for their comment. We completely understand that the paper is lengthy and have revised and modified it considerably, such as reducing the Introduction and Variable Description sections to shorten the paper. However, we believe that figures are very crucial to our analysis (particularly the marginal effect analysis with interaction effect), as they allow us to examine whether the effect of social groups on anthropometric outcome changes with the change in other SES indicators such as wealth index, maternal education level, and geographical location (urban vs. rural). Notably, the figures make it easy to interpret the magnitude of the effect of the caste on the anthropometric failures rather than the direction of changes. Therefore, we have decided to keep the figures in the revised manuscript. 

With sincere thanks to Editor and Reviewers, we would also like to mention that we shortened the title of our manuscript from ‘Understanding the impact of Ethnicity/Caste and Child Anthropometric Outcomes in India using the National Family Heath Survey 2015-16 and 2019-21’ to ‘Ethnicity/Caste and Child Anthropometric Outcomes in India using the National Family Heath Survey 2015-16 and 2019-21.’

Again, we thank the Editor and Reviewers for their kind consideration of our manuscript and for giving us the opportunity to submit the revised manuscript.

---

## [Decision Letter · Decision Letter 1]

13 Sep 2024

Ethnicity/Caste and Child Anthropometric Outcomes in India using the National Family Heath Survey 2015-16 and 2019-21

PONE-D-24-16005R1

Dear Dr. Sakshi Pandey,

We’re pleased to inform you that your manuscript has been judged scientifically suitable for publication and will be formally accepted for publication once it meets all outstanding technical requirements.

Kind regards,

Prem Shankar Mishra

Academic Editor

PLOS ONE

Additional Editor Comments (optional):

The authors have incorporated the comments and rationally responded to reveiwers comments. 

Reviewers' comments:

Reviewer's Responses to Questions

**Comments to the Author**

1. If the authors have adequately addressed your comments raised in a previous round of review and you feel that this manuscript is now acceptable for publication, you may indicate that here to bypass the “Comments to the Author” section, enter your conflict of interest statement in the “Confidential to Editor” section, and submit your "Accept" recommendation.

Reviewer #1: All comments have been addressed

Reviewer #2: All comments have been addressed

2. Is the manuscript technically sound, and do the data support the conclusions?

Reviewer #1: Yes

Reviewer #2: Yes

3. Has the statistical analysis been performed appropriately and rigorously? 

Reviewer #1: Yes

Reviewer #2: Yes

4. Have the authors made all data underlying the findings in their manuscript fully available?

Reviewer #1: Yes

Reviewer #2: Yes

5. Is the manuscript presented in an intelligible fashion and written in standard English?

Reviewer #1: Yes

Reviewer #2: Yes

6. Review Comments to the Author

Reviewer #1: I have reviewed your manuscript and satisfied with corrections that have been made. I recommend a thorough proofreading before final submission.

Reviewer #2: The Variance Inflation Factors seem to be too low. The authors are asked to check it again.

I still maintain that the figures are not very much necessary. However, I don’t insist that the figures should be deleted.

7. PLOS authors have the option to publish the peer review history of their article (what does this mean?). If published, this will include your full peer review and any attached files.

Reviewer #1: No

Reviewer #2: No

---

## [Editor Report · Acceptance letter]

23 Sep 2024

PONE-D-24-16005R1 

PLOS ONE

Dear Dr. Pandey, 

I'm pleased to inform you that your manuscript has been deemed suitable for publication in PLOS ONE. Congratulations! Your manuscript is now being handed over to our production team.

Kind regards, 

on behalf of

Dr. Prem Shankar Mishra 

Academic Editor

PLOS ONE